# Foot Problems and Their Associations with Toe Grip Strength and Walking Speed in Community-Dwelling Older Individuals Using Day Services: A Cross-Sectional Study

**Kashiko Fujii [1,\*], Atsuko Maekawa [2], Takuyuki Komoda [3], Nozomi Kawabe [4], Ryouhei Nishimura [5], Yasunori Sakakibara [6], Takahiko Fukumoto [7] and Minna Stolt [8,9]**

1   Nursing Department, Tokyo Kasei University, 2-15-1, Inaniyama, Sayama City 350-1398, Japan
2   Ex Graduate School of Medicine, Department of Nursing Science, Nagoya University, 1-1-20 Daiko-Minami, Higashi-ku, Nagoya City 461-8673, Japan
3   Toyohashi Heart Center Plastic Surgery, Gifu Heart Center, 4-14-4 Yabuta Minami, Gifu City 500-8384, Japan
4   Division of Host Defense Sciences, Department of Integrated Health Sciences, Graduate School of Medicine, Nagoya University, 1-1-20 Daiko-Minami, Higashi-ku, Nagoya City 461-8673, Japan
5   Surgical Intensive Care Unit, Department of Nursing, Nagoya University Hospital, 65 Turumai-cho, Shouwa-ku, Nagoya City 466-8560, Japan
6   The Unit of Neurosurgery and Orthopedic Surgery, Department of Nursing, Nishichita General Hospital, 3-1-1 Nakanoike, Tokai City 477-8522, Japan
7   Department of Physical Therapy, Health Science, Kio University, 4 Chome-2-2 Umaminaka, Koryo, Kitakatsuragi District, Nara 635-0832, Japan
8   Department of Nursing Science, University of Turku, FI-20014 Turku, Finland
9   Department of Nursing Science, University of Eastern Finland, PL 1627, FI-70211 Kuopio, Finland
*   Correspondence: fnhiroaki45@gmail.com or fujii-k@tokyo-kasei.ac.jp

**Abstract:** Foot disorders in older individuals compromise balance and contribute to postural and gait instabilities, causing a decrease in the activities of daily living and quality of life. In this cross-sectional study, we analyzed the foot-related data of 160 frail older participants who attended day service centers in A prefecture in Japan to determine the prevalence of foot problems and their associations with toe grip strength and walking speed in frail older people. Multiple regression analysis was used to identify foot-related variables that correlated with toe grip strength and walking speed. The prevalence rates of skin dryness (Support level 88.2%, Care level 85.2% for men; Support level 84.9%, Care level 93% for women) and suspected and existing fungal infections in nails (Support level 94.1%, Care level 92.6% for men; Support level 98.1%, Care level 95.2% for women) were high in both sexes regardless of the level of care required. Furthermore, in both sexes, the prevalence rates of toe and arch deformities were significantly increased in the people who required care. Regression analysis revealed that some right-sided foot-related problems were significantly associated with right toe grip strength and walking speed. The decrease in toe grip strength was significantly associated with walking speed. Our study provides evidence that some specific conditions were associated with toe grip force and walking speed. This finding can contribute to future strategies to protect foot health in community-dwelling older individuals.

**Keywords:** community-dwelling older people; foot problem; toe grip force; walking speed

## 1. Introduction

Foot problems in community-dwelling older people have become a growing concern worldwide, with studies reporting that 75–80% of community-dwelling people >65 years of age have more than one foot problem [1,2] and frequently complain of foot pain [3]. Foot problems are especially prevalent among frail older individuals owing to the difficulties that they face while taking care of their feet [1,4]. Major foot problems include toenail disorders, lesser toe deformities, arch deformity, corns and calluses, maceration between the

toes, skin dryness, edema, and hallux valgus [5–7]. Aging causes morphological features and structural and functional changes to maintain postural stability and may lead to an increase in the incidence of falls [8]. Frailty is defined in various studies [9–11]. According to a statement from the Japan Geriatrics Society, the concept of frail is "a state of increased vulnerability to stress. A state of being prone to turning points such as life dysfunction, need for nursing care, and death" [12].

The subjects of this study were considered frail older people. They applied for long-term care insurance and to be certified to receive the benefit of long-term insurance. They received services including attending adult day service centers, although the number and content of services received varied depending on categories.

In Japan, the older population is rapidly increasing; the rate of change in the aging population is 28%, compared with 9.03% globally [13]. With the increasing number of older people, fall injuries are also a concern in Japan. Fall injuries accounted for roughly 80% of accidents in the daily life of elderly individuals in 2016 [14]. The number of deaths due to fall-related injuries has increased from 5944 in 2016 to 8803 in 2018 [15]. Previous studies showed that foot problems are associated with falls [1,2,16]. Given that the frequency of falls is high in Japan, foot problems should be seriously considered. However, awareness about the impact of foot problems on the older population in Japan is lacking [17,18].

Although foot studies in community populations are limited, unique studies related to toes have been conducted in Japan [19,20]. Two types of measurement devices, which are those for toe-gap force measurement [21] and toe grip dynamometers [22], were developed to assess toe force in relation to dynamic balance or functional mobility, although they target different muscles [20].

A previous study used these devices to investigate the association between the toes and the lower limb muscles that maintain balance [21,23]. Since the toe movement is a product of the collaboration of flexor–tensor muscles, which compose the lower part of the foot [24], toe stability provides postural stability during walking. Based on the assumption that toe exercise might relate to the dorsal and ventral premotor cortexes, the same as the finger movement, the possibility of an association between toe exercise and cognitive function was also reported [25]. Another device, Foot Look (Foot Look Inc., City of Fukuoka, Japan), was developed to measure plantar foot pressure and structure, although this device is not yet well known worldwide. The study focuses on foot problems in community-dwelling older people in Japan and targeted items such as toe function and plantar foot pressure; however, the foot should be analyzed from various perspectives to evaluate foot health. For foot health, walking speed is inevitable as an indicator of foot functioning. Therefore, walking speed has been studied in various aspects [26–30].

It is necessary to evaluate the existing condition of the foot among older people and explore the future strategy to protect foot health. The aims of this study were: (1) to determine the prevalence of foot conditions in older people in the community and (2) to examine the associations between foot conditions and toe grip strength and walking speed.

## 2. Materials and Methods

### 2.1. Design

This is a cross-sectional study using random cluster sampling of community-dwelling people aged >65 years who attended daycare or day rehabilitation centers in A prefecture in Japan. Data were collected from July in 2019 to October in 2019.

### 2.2. Sampling and Participants

In Japan, there are two types of insurance for the healthcare of older people: medical and long-term care insurance, which is a compulsory public program run by the municipal government and specific districts in Tokyo. Those insured under the long-term insurance system include category 1 (people aged 65 years or older) and 2 (people aged 40–64 years who are covered by a health insurance program).

Applicants for care services must be screened. To use a long-term health insurance system, older people in Japan must apply and obtain certification by completing certain processes, including screening. Certified older people are categorized into seven levels, labeled support levels 1 and 2 and care levels 1–5, to receive services, depending on the degree of care they need [31]. Applicants in a support level can do most of the basic activities in daily life by themselves, but they need nursing care up to some extent. Applicants in a care level need some kind of assistance to do certain activities in daily life because they are not able to do them by themselves. The services covered by long-term care insurance are divided into home-based, community-based, and facility-based services [32]. In home-based services, day service and daycare centers provide daily care, such as hygiene, meals, recreation, and physical rehabilitation, depending on the participant's plan.

For those who were categorized between care level 1 and 5 (excepting support level 1 and 2), 1,159,200 people used the day service, whereas 436,600 used the day care service offering rehabilitation [33].

According to the sample size calculator [34], 385 samples were required, and the 95% confidence interval and 5% margin of error were calculated. The sample size was also calculated using G*Power to validate the sample size. G*Power is well-known power analysis program for statistical tests [35]. Following Cohen's parameter for the effect size for linear multiple regression [36] and fixed mode and $R^2$ deviation from zero, we input the effect size for $f^2$ (medium = 0.15), $\alpha$ = 0.05, power = 0.8, and the number of predictors = 2, which are toe grip strength and walking speed. As a result of the formula, the sample size was 68.

The coefficient of determination $R^2$ measures the proportion of variance explained by a statistical model which predicts an outcome [37] and Cohen's $f^2$ is for calculating the effect size measure for linear regression [38].

We predicted that older adults would be more likely to drop out of the study even after participating. In addition, since this study is unprecedented in Japan, we predicted that there would be few facilities that would cooperate in participating in the study. Therefore, we aimed for a sampling number of 385, which was much larger than the sampling number calculated by G Power.

Therefore, although we aimed to recruit 385 individuals, a total sample size of 160 was used to analyze the data.

We randomly selected 400 one-day service centers in A prefecture in Japan. The number of attendants varied depending on the center. A letter of invitation was sent to the randomly-selected facilities in A prefecture in Japan to participate in the study. Some invitations returned due to address changes. Of the 27 facilities that responded, 20 facilities agreed to participate in the study. The manager or staff members of each facility were willing to explain the study to the participants and their families beforehand. As the survey time was limited for each facility, the study participants were selected by the facilities. The examiner explained the contents of the study to all participants and obtained oral and written consent from all participants. The oral and written consent procedure was approved by the ethics committee of Nagoya University. In the case that the institutional staff considered the need for agreement from the participant's family to supplement the participant's understanding of the research, they asked their families to sign the necessary agreements.

The inclusion criteria called for people aged >65 years who were receiving services through long-term care insurance and could walk with or without a walking aid. Participants with foot infections, such as ulcers or wounds, were excluded.

### 2.3. Study Instruments

We used assessment methods and three types of measuring devices to obtain foot data on structural and functional aspects, as follows: a Foot Look (Figure 1), a toe grip machine (Figure 2), and a thermometer. The main author developed a two-page recording sheet based on previous studies to record data on the study items [18,39]. The recording

sheets were evaluated by two independent researchers before data collection. Table 1 presents the definitions, measurement units, and assessment methods of foot-related items for assessment and measurement of the item selection of the recording sheet (https://footlook.co.jp/, accessed on 1 May 2019). A Foot Look (Foot Look Inc.) was used to measure foot length, foot width, and toe angle from the sole. The device uses a computer and scanner to provide visual images of the soles, including the hallux valgus and floating toe, and the weight distribution of the body to the floor (Figure 1: https://footlook.co.jp/, accessed on 1 May 2019). We also used a toe grip dynamometer to measure toe flexor strength, which indicates toe performance (Figure 2: T.K.K.3364 Takei Scientific Instrument; https://www.takei-si.co.jp/products/719/, accessed on1 May 2019).

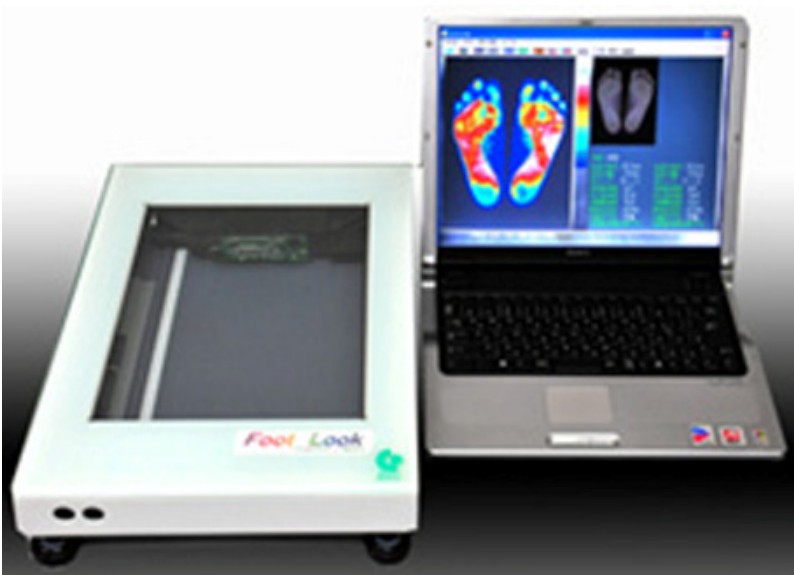

**Figure 1.** Foot Look.

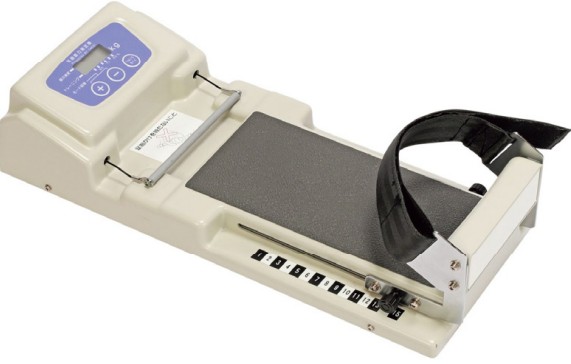

**Figure 2.** Toe grip dynamometer.

### 2.4. Measurement

Foot-related items were classified as those requiring participants to perform either active or passive actions at the time of foot measurement. Passive actions were defined as instances when the examiner assessed the participant in a seated position with no resistance. Actions were considered active when the participants were asked to move their toes or feet or to walk or step on the measurement board of the machine.

**Table 1.** Measurement items.

| Item | Definition of Measurement | Measurement | Scale Definition for This Study's Analysis | Related References (Examples) |
|---|---|---|---|---|
| Arch deformities | Abnormal arch (e.g., high arch, flatfoot) | Examiner assessed by visualization in the following cases: Wider foot width with broken lateral arch Flat feet with a low arch Foot with a high arch | Exists/does not exist | Flatfoot [39,40] |
| Ingrown toenails | Corners of nails growing into the skin | Examiner's assessment | Exists/does not exist | Ingrown toenails [40] |
| Sensitivity of toe (1st, 3rd, and 5th) | Foot sensation; usually, if the subject feels numbness in two or more than three toes, it is considered abnormal | Examiner touched the participant's toes and asked whether they could feel it (Ipswich touch test) | Participants have/do not have sensation (if the participant did not feel, then sensation was impaired) | Ipswich touch test [41,42] |
| Foot circulation | Palpable posterior tibial arteries | Examiner palpated the pulsing artery halfway between the medial malleolus posterior border and the Achilles tendon | Examiner was able to palpate the pulse or was not able to palpate it | Posterior tibial arteries [40] |
| Edema | Pitting edema | The skin was pressed over the tibia, and the compressed area was examined after the release of pressure: +1 for a barely detectable indentation, +2 for a slight indentation (15 s to rebound), +3 for a deeper indentation, +4 for a deeper indentation that takes more than 30 s to rebound | Above +2 was counted as edema in this study | [43] |
| Skin color | Skin color (red, white, purple, etc.) | Examiner's assessment | Normal or abnormal skin color (red, purple, white, etc.) | [44] |
| Toe deformities | - | Existence of mallet toes, hammer toes, claw toes, hallux valgus, and lesser toe deformities | Number of deformed toes | [5] |
| Skin lesions | Hyperkeratotic skin lesions (area of thickened skin caused by repeated friction or pressure). Corns are usually hard and painful when pushed | Examiner's assessment | Number of skin lesions | [1] |
| Maceration between toes | Peeling and fissuring between the toes, lighter in color, sometimes white | Examiner's assessment | Number of instances | [6,45] |

**Table 1.** *Cont.*

| Item | Definition of Measurement | Measurement | Scale Definition for This Study's Analysis | Related References (Examples) |
|---|---|---|---|---|
| Nail color change | Lack of nail color such as white or yellow | Examiner's assessment | Number of affected toenails | [46] |
| Long nails | Beyond the tip of the toe | Examiner's assessment | Number of affected toenails | NA |
| Thickened nail | Nail is excessively thick | Examiner's assessment | Number of toenails >2 mm in thickness | [5] |
| Skin dryness | Corneal layer water loss | Overall dry skin score (ODS *) EEMCO guideline Note: Specified symptom sum score excluded in this study | Exist or does not exist, then classified into a score of 0–4, 0 means no exist | [47,48] |
| Suspected or existing nail fungal infection | Already diagnosed or suspected from nail color | Examiner's assessment | Number of affected toenails | [49] |
| Walking speed | 4 m walking speed | Measured by examiners or research assistants | Seconds | [50] |
| Hallux valgus | Degree of hallux valgus | Examiners' assessment was based on Foot Look (machine) images. The hallux valgus angle was categorized into HVA $\leq$ 15°, 15° < HVA $\leq$ 20°, 20° < HVA $\leq$ 40°, and HVA > 40° | Degrees | [51] |
| Toe-spreading width ** | Toe muscle weakness and decreased range of motion | Examiners measured the width using tape. The distance between the first and second toe was measured midway along their length, and the same was done between the second and fifth toe. The subject was then asked to spread the toes as far as possible, and the difference was calculated. In this research, the distance between the first and second toes was measured.  | cm | [52] |

**Table 1.** *Cont.*

| Item | Definition of Measurement | Measurement | Scale Definition for This Study's Analysis | Related References (Examples) |
|---|---|---|---|---|
| Toe grip strength | The toe strength reading was displayed on the machine when a participant gripped the bar using the toes at maximal force for 3 s | Examiners or research assistants measured with a toe grip dynamometer (T.K.K.3364 Takei Scientific Instruments Co., Ltd.) Unit of toe grip strength was kg | kg | [53] |
| Floating toes | Toes that are not completely in contact with the ground | Examiners assessed based on Foot Look (machine) images | Number of toes that are not completely in contact with the ground | [54,55] |

* Overall dry skin score (ODS): A scoring scale combining all major and minor signs of dry skin (xerosis), as follows [56]: 0: Absent; 1: Faint scaling, faint roughness, and dull appearance; 2: Small scales in combination with a few larger scales, slight roughness, and whitish appearance; 3: Small and larger scales uniformly distributed, definite roughness, possibly slight redness, and possibly a few superficial cracks; 4: Dominated by large scales, advanced roughness, redness present, eczematous changes, and cracks. https://www.takei-si.co.jp/products/719/, accessed on 1 May2019; https://footlook.co.jp/, accessed on 1 May 2019; ** The measurement of toe-spreading width was developed by the main author based on her clinical experiences.

The measurement of walking speed was considered active. The participants were carefully monitored to prevent falls or injury during the assessments and data collection.

*2.5. Equipment*

2.5.1. Foot Look

When the subject stands on the scanner and places the foot sole and all five toes on the scanner, the image of the sole is displayed on the computer in color differences.

By inserting four lines into the sole image by using a function, the computer automatically calculated the angle of the thumb (big toe angle), the angle of the little toe, and the open toe angle (FICK angle). The display also showed the length of the sole (cm), width of the sole (cm), shoe width (A~G), ground contact ratio, and degree of development of the ball of the foot.

As shown in Figure 1, the toes touching the floor can be observed from the display of Foot Look, and whether the hallux valgus is present and whether a difference exists between the left and right feet in how they contact the floor can be determined. Any toe that was not completely touching the floor on the displayed image was considered a floating toe.

2.5.2. Toe Grip Machine

The participants were asked to sit upright, with both the hip and knee joints at 90° and the ankle joints in a neutral position. While the heel was fixed in place with a stopper, the grip bar was positioned using the first proximal phalanx as a landmark. The participants gripped the bar with the greatest possible force for 3 s using their toes [53,54], and the force measurement was recorded.

2.5.3. Thermometer

In our clinical experience, the feet of many older people are cool to the touch. To supplement the data for blood circulation, the skin temperature of the foot was also measured.

*2.6. Study Preparation*

Nine research assistants were educated and trained to gain knowledge of the human foot and practiced skills for measuring foot-related variables. A physiotherapist who researches toes provided lectures on basic knowledge of the anatomy and physiology of the foot. They also learned how to operate foot grip measurement tools and the Foot Look and on interview methods.

*2.7. Study Setting*

The author coordinated the time with each facility, and the author and 2~3 research assistants visited the facility on the designated day for measurements. The foot assessment was done by an author with a certificate of Fusspflege. Foot measurements and gait measurements were mainly performed by research aides trained under the author.

*2.8. Ethics*

The research conformed to the Declaration of Helsinki, 2013. We obtained informed consent from all participants with the approval of the facility providers and some of the families. Furthermore, the study was approved by the ethics committee of Nagoya University (approval number: 2019–0150).

*2.9. Data Analysis*

Those insured under the long-term insurance system include category 1 (people aged 65 years or older) and 2 (people aged 40–64 years who are covered by a health insurance program).

Applicants in a support level can do most of the basic activities in daily life by themselves, but they need nursing care up to some extent. Applicants in a care level need some

kind of assistance to do certain activities in daily life because they are not able to do them by themselves. Since the degrees of assistance are different in the two levels, we compared the participants in the support level and care level.

Additionally, since women and men have different muscle mass, we conducted data analysis by dividing men and women [57].

Descriptive statistics were used to analyze the demographic characteristics and foot-related assessment and measurement data. Sequentially, we calculated the mean, standard deviation, median, maximum, and minimum values, and employed a paired *t*-test, Fisher's exact test, or Mann–Whitney U test. Fisher's exact test is used when the denominator is the "applicable %" and the denominator is the "not applicable %" which can be replaced by a $2 \times 2$ cross table. Mann–Whitney's U test is used for the data when it is doubtful whether the distribution is normal or not.

Cumulative methods for multiple foot problems or dichotomous scales were based on the presence or absence of foot disorders as described in a previous study [15,58].

We compared the data by gender and level of care, which had two subcategories: (1) the support level, at which patients are relatively independent in daily life, and (2) the care level, at which patients need help from others through their long-term care insurance. Most items were analyzed by categorizing the right and left feet separately. Some of the participants could not move their feet sufficiently because of cerebral impediments or back problems.

Multiple regression analysis was used to identify foot-related variables that correlated with toe grip strength and walking speed.

## 3. Results

The total number of participants was 160, as shown in the flow diagram of participants in Figure 3. Some missing data were attributed to 10 patients' inability to step up onto the table of the Foot Look machine because of physical limitations or their inability to grip the bar on the toe grip dynamometer. Some other data could not be obtained because of one-sided paralysis caused by cerebrovascular or hip problems or other issues. Furthermore, one participant could not participate in measurements that required active movement such as spreading the toes. Therefore, the number of participants in the analysis accordingly varied. The sample number was valid for multiple regression analysis, as previously mentioned.

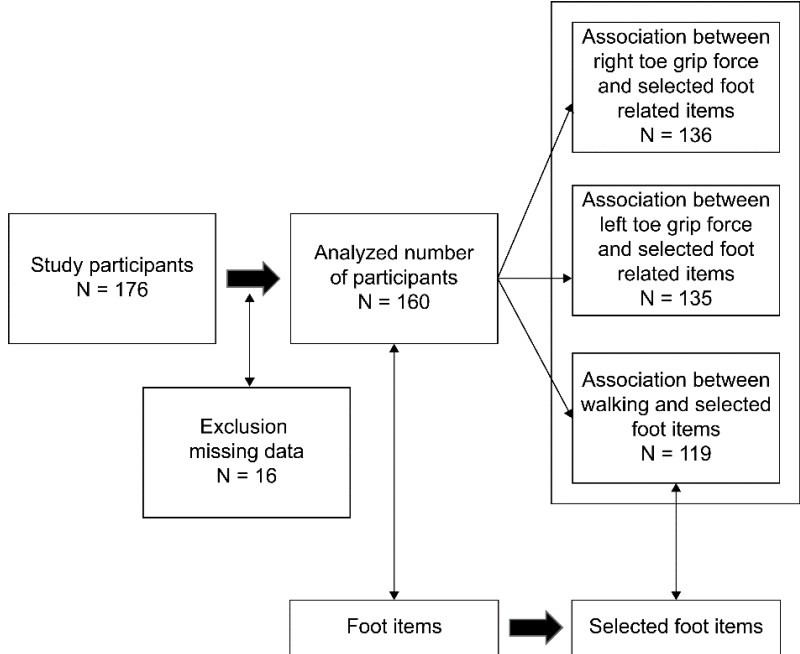

**Figure 3.** Flow diagram of participants.

The demographic characteristics of the participants, including age, Barthel Index, and history of the disease, are presented in Table 2. Out of 160 analyzed participants, the majority were women (116, 72.5%); the average age of the participants was 82 years for men and 84 years for women. Additionally, 17 (39%) and 27 (61%) men and 53 (46%) and 63 (54%) women were classified into the support and care levels, respectively. The most prevalent diseases were hypertension (41.9%), brain disease (history of cerebrovascular accidents) (19.4%), and diabetes (16.3%) (Table 3). When gender and medical history were compared by level of care, it was noted that men at the support level had a higher prevalence of diabetes (41.2%). At the care level, the prevalence of cerebrovascular accidents in men was 59.3%, and hypertension was present in 48.1% of men and 34.9% of women.

The results of the 4 m walk test, skin temperature, toe grip force, and toe-spreading width are shown in Table 3.

Those insured under the long-term insurance system include categories 1 (people aged ≥65 years) and 2 (people aged 40–64 years who are covered by a health insurance program). Applicants for care services had to be screened. Accepted applicants were categorized into support levels 1 and 2 and care levels 1–5.

Applicants at the support level can perform most of the basic activities of daily living (ADLs) by themselves, but they need nursing care to a certain extent. Applicants at the care level need some form of assistance to perform certain ADLs because they cannot perform them by themselves, although it is possible for them to complete most of the basic ADLs.

The degree of hallux valgus was measured with the Foot Look (Foot Look Inc.). Toe grip strength was measured with a toe grip dynamometer (T.K.K.3364., Takei Scientific Instrument).

The mean times to complete the walking speed test were higher at the care level for both men and women (6.0 ± 2.2 vs. 8.6 ± 4.3 s for men and 5.6 ± 1.9 vs. 7.9 ± 3.8 s for women at the support and care levels, respectively), and the right and left toe-spreading widths were significantly narrower at the support level than at the care level for men and women (0.005 and 0.007, respectively). Both the right and left toe grip strength was significantly higher for men at the support than at the care level. The skin temperature on the lower extremities ranged from 33.6 °C to 34.2 °C.

Table 4 presents the foot assessment results. Significant differences were observed only in arch deformity between women at the care level and those at the support level ($p = 0.037$). Skin dryness and suspected or existing nail fungal infection were highly prevalent in both men and women in general (skin dryness occurrence was above 85.2% and 77.8% on each side for men and women, respectively; suspected or existing nail fungal infection occurrence was above 92.6% and 95.2% for men and women, respectively). The prevalence rates of nail color change (above 69.8%) and arch deformity (63%) were relatively high in both sexes. However, the prevalence of nail color change was higher in men, whereas that of the arch deformity was higher in women. Corns and calluses had a relatively low prevalence in both sexes. The weight distribution rate on the floor was higher on the right than on the left side. Subjects with more deformities on the right side could likely maintain their right foot longer on the floor and apply a higher pressure on it due to lower mobility and fewer motor skills on the right side. The prevalence of hallux valgus exceeding 15° was higher in women than in men (8.3–26.7% and 28.8–43.1%, respectively).

Table 5 presents the foot problems of participants with diabetes. A higher rate of participants had floating toes and toe deformities (73.1% and 69.2%), arch deformities (84.6% and 76.9%), toe deformities (80.8% and 73.1%), thickened nails (61.5% and 65.4%), and suspected or existing nail fungal infection (96.2% and 96.2%).

**Table 2.** Sample Characteristics.

| | | Men | | | | | | | | Women | | | | | |
| | Support level | | | Care level | | | | | Support level | | | Care level | | | |
| | n | Mean | SD | n | Mean | SD | *p*-value | | n | Mean | SD | n | Mean | SD | *p*-value | |
| Age | 17 | 83.4 | 4.9 | 27 | 80.4 | 7.1 | 0.137 | | 53 | 83.2 | 6.3 | 63 | 85.5 | 6.1 | 0.049 | |

| Item | Support level (n = 17) | | | | Care level (n = 27) | | | | | Support level (n = 53) | | | | Care level (n = 63) | | | | |
| | n | Median | 25% | 75% | n | Median | 25% | 75% | *p*-value | n | Median | 25% | 75% | n | Median | 25% | 75% | *p*-value |
| Barthel Index | 17 | 95 | 90 | 100 | 24 | 85 | 71.3 | 100 | 0.071 | 52 | 95 | 90 | 100 | 62 | 95 | 85 | 100 | 0.89 |

| | Men | | | | | Women | | | | |
| | Support level | | Care level | | | Support level | | Care level | | |
| Item | n | Rate | n | Rate | *p*-value | n | Rate | n | Rate | *p*-value |
| Diabetes | 7 | 41.20% | 5 | 18.50% | 0.164 | 7 | 13.20% | 7 | 11.10% | 0.78 |
| Hypertension | 8 | 47.10% | 13 | 48.10% | 1 | 24 | 45.30% | 22 | 34.90% | 0.341 |
| Kidney disease | 1 | 5.90% | 1 | 3.70% | 1 | 1 | 1.90% | 1 | 1.60% | 1 |
| Arteriosclerosis obliterans | 0 | 0% | 3 | 11.10% | 0.272 | 1 | 1.90% | 0 | 0.00% | 0.457 |
| Heart disease | 3 | 17.60% | 4 | 14.80% | 1 | 11 | 20.80% | 14 | 22.20% | 1 |
| Brain disease | 2 | 11.80% | 16 | 59.30% | 0.002** | 4 | 7.50% | 9 | 14.30% | 0.377 |
| Rheumatism | 0 | 0% | 1 | 3.70% | 1 | 3 | 5.70% | 0 | 0.00% | 0.092 |
| Hyperlipidaemia | 0 | 0% | 1 | 3.70% | 1 | 3 | 5.70% | 6 | 9.50% | 0.506 |
| Lung disease | 0 | 0% | 1 | 3.70% | 1 | 4 | 7.50% | 0 | 0.00% | 0.041* |
| Incontinence | 1 | 5.90% | 2 | 7.40% | 1 | 0 | 0.00% | 4 | 6.30% | 0.124 |

Note: We were not able to obtain data from all participants. For the Barthel Index: Mann–Whitney U Test, * $p < 0.05$; ** $p < 0.01$; data are median, 25%, and 75%; the Barthel Index is a simple assessment and evaluation method to measure the 10 types of movement required for daily life. Some participants were not able to answer the questions for the Barthel Index. For the history of disease: N (%) represents the number of participants who stated which disease(s) they have or had (multiple answers were permitted); data are median, 25%, and 75%.

**Table 3.** Four m walking speed, skin temperature, right toe grip strength, left toe grip strength.

| | | Men | | | | | | | | Women | | | | | |
| | | Support Level | | | Care Level | | | | | Support Level | | | Care Level | | |
| | n | Mean | SD | n | Mean | SD | *p*-value | n | Mean | SD | n | Mean | SD | *p*-value |
|---|---|---|---|---|---|---|---|---|---|---|---|---|---|---|
| 4-meter walking speed (second) | 16 | 6.00 | 2.20 | 21.00 | 8.60 | 4.30 | 0.039 * | 48 | 5.6 | 1.9 | 51 | 7.9 | 3.8 | 0.000 ** |
| Skin temperature (foot) | 15 | 33.60 | 1.40 | 23.00 | 33.60 | 2.10 | 0.998 | 47 | 34.2 | 1.7 | 58 | 34.2 | 1.8 | 0.944 |
| Right toe grip strength | 16 | 7.00 | 5.10 | 26.00 | 4.30 | 3.50 | 0.048 * | 52 | 4 | 2.7 | 62 | 3.6 | 3.6 | 0.509 |
| Left toe grip strength | 16 | 6.30 | 3.20 | 26.00 | 4.10 | 3.20 | 0.038 * | 52 | 3.8 | 2.3 | 62 | 3.5 | 2.5 | 0.588 |

| | | Men | | | | | | | | Women | | | | | | | |
| Item | | Support level (n = 17) | | | Care level (n = 27) | | | | | Support level (n = 53) | | | Care level (n = 63) | | | |
| | n | Median | 25% | 75% | n | Median | 25% | 75% | *p*-value | n | Median | 25% | 75% | n | Median | 25% | 75% | *p*-value |
|---|---|---|---|---|---|---|---|---|---|---|---|---|---|---|---|---|---|---|
| Widths of opening toes (1st and 2nd), right foot | 17 | 1 | 0.4 | 2.35 | 27 | 0.3 | 0.2 | 0.7 | 0.014 * | 52 | 0.6 | 0.2 | 1.7 | 62 | 0.3 | 0.1 | 1 | 0.005 ** |
| Widths of opening toes (1st and 2nd), left foot | 17 | 0.6 | 0.25 | 2.15 | 27 | 0.3 | 0.2 | 0.5 | 0.048 * | 53 | 0.6 | 0.2 | 1.8 | 62 | 0.2 | 0.1 | 0.5 | 0.007 ** |

Walking speed, Toe grip strength, foot skin temperature, and width of opening toes: for walking speed, toe grip strength, and foot skin temperature: Paired *t*-test, * *p* < 0.05; ** *p* < 0.01; Toe grip strength was measured with a toe grip dynamometer (T.K.K.3364. Takei Scientific Instrument); for toe spreading width: Mann–Whitney U Test, * *p* < 0.05; ** *p* < 0.01; data are presented as the median, 25%, and 75%; the toe spreading width was measured as the change in distance between the 2nd and 5th toe halfway along their length. The examiner asked participants to spread their toes to maximum. The measurement of toe-spreading width was developed by the main author based on her clinical experiences. In some participants, the toe-spreading width could not be tested.

**Table 4.** Foot assessment.

| | | Men | | | | | Women | | | | |
| | | Support Level (n = 17) | | Care Level (n = 27) | | | Support Level (n = 53) | | Care Level (n = 63) | | |
| | Item | n | Rate | n | Rate | *p*-Value | n | Rate | n | Rate | *p*-Value |
|---|---|---|---|---|---|---|---|---|---|---|---|
| Right foot | Floating toe | 9 | 39.1% | 14 | 60.9% | 1.000 | 37 | 47.4% | 41 | 52.6% | 0.841 |
| | Toe deformities | 8 | 47.1% | 23 | 85.2% | 0.015 * | 46 | 86.8% | 54 | 85.7% | 1.000 |
| | Skin lesions (corns and calluses) | 1 | 5.9% | 3 | 11.1% | 1.000 | 11 | 20.8% | 7 | 11.1% | 0.200 |
| | Maceration between toes | 1 | 5.9% | 2 | 7.4% | 1.000 | 2 | 3.8% | 6 | 9.5% | 0.287 |
| | Nail color change | 15 | 88.2% | 23 | 85.2% | 1.000 | 42 | 79.2% | 54 | 85.7% | 0.461 |
| | Arch deformities | 12 | 70.6% | 20 | 74.1% | 1.000 | 44 | 83.0% | 60 | 95.2% | 0.037 * |
| | Long nails | 11 | 64.7% | 18 | 66.7% | 1.000 | 40 | 75.5% | 53 | 84.1% | 0.255 |
| | Thickened nails | 10 | 58.8% | 17 | 63.0% | 1.000 | 32 | 60.4% | 42 | 66.7% | 0.562 |
| | Ingrown nails | 4 | 23.5% | 5 | 18.5% | 0.716 | 19 | 35.8% | 17 | 27.0% | 0.321 |
| | Skin dryness | 15 | 88.2% | 23 | 85.2% | 1.000 | 45 | 84.9% | 59 | 93.7% | 0.139 |
| | Edema | 16 | 94.1% | 27 | 100.0% | 0.386 | 50 | 94.3% | 61 | 96.8% | 0.659 |
| | Skin color change | 17 | 100.0% | 27 | 100.0% | 1.000 | 51 | 96.2% | 63 | 100.0% | 0.207 |
| | Suspected or existing nail fungal infection | 16 | 94.1% | 25 | 92.6% | 1.000 | 52 | 98.1% | 60 | 95.2% | 0.624 |
| | Hallux valgus angle | | | | | | | | | | |
| | HVA ≤ 15° | 14 | 87.5% | 22 | 91.7% | | 34 | 64.2% | 40 | 70.2% | |
| | 15° < HVA ≤ 20° | 1 | 6.3% | 2 | 8.3% | 0.455 | 8 | 15.1% | 7 | 12.3% | 0.863 |
| | 20° < HVA ≤ 40° | 1 | 6.3% | 0 | 0% | | 9 | 17.0% | 9 | 15.8% | |
| | HVA > 40° | 0 | 0% | 0 | 0% | | 2 | 3.8% | 1 | 1.8% | |
| Left foot | Floating toe | 8 | 33.3% | 16 | 66.7% | 1.000 | 39 | 45.9% | 46 | 54.1% | 1.000 |
| | Toe deformities | 10 | 58.8% | 18 | 66.7% | 0.749 | 40 | 75.5% | 55 | 87.3% | 0.146 |
| | Skin lesions (corns and calluses) | 1 | 5.9% | 1 | 3.7% | 1.000 | 12 | 22.6% | 12 | 19.0% | 0.653 |
| | Maceration between toes | 1 | 5.9% | 1 | 3.7% | 1.000 | 1 | 1.9% | 5 | 7.9% | 0.217 |
| | Nail color change | 14 | 82.4% | 22 | 81.5% | 1.000 | 37 | 69.8% | 50 | 79.4% | 0.284 |
| | Arch deformities | 11 | 64.7% | 17 | 63.0% | 1.000 | 42 | 79.2% | 54 | 85.7% | 0.461 |
| | Long nails | 10 | 58.8% | 20 | 74.1% | 0.334 | 36 | 67.9% | 47 | 74.6% | 0.536 |
| | Thickened nails | 10 | 58.8% | 16 | 59.3% | 1.000 | 24 | 45.3% | 36 | 57.1% | 0.263 |
| | Ingrown nails | 5 | 29.4% | 12 | 44.4% | 0.360 | 21 | 39.6% | 25 | 39.7% | 1.000 |
| | Skin dryness | 17 | 100.0% | 25 | 92.6% | 0.515 | 46 | 86.8% | 49 | 77.8% | 0.235 |
| | Edema | 8 | 47.1% | 16 | 59.3% | 0.539 | 21 | 39.6% | 30 | 47.6% | 0.454 |
| | Skin color change | 13 | 76.5% | 18 | 66.7% | 0.735 | 43 | 81.1% | 41 | 65.1% | 0.063 |
| | Suspected or existing nail fungal infection | 16 | 94.1% | 25 | 92.6% | 1.000 | 52 | 98.1% | 60 | 95.2% | 0.624 |
| | Hallux valgus angle | | | | | | | | | | |
| | HVA ≤ 15° | 11 | 73.3% | 21 | 87.5% | | 37 | 71.2% | 33 | 56.9% | |
| | 15° < HVA ≤ 20° | 2 | 13.3% | 2 | 8.3% | 0.482 | 2 | 3.8% | 10 | 17.2% | 0.086 |
| | 20° < HVA ≤ 40° | 2 | 13.3% | 1 | 4.2% | | 12 | 23.1% | 15 | 25.9% | |
| | HVA > 40° | 0 | 0% | 0 | 0% | | 1 | 1.9% | 0 | 0% | |
| Weight distribution | Left | 5 | 31.3% | 11 | 47.8% | | 17 | 34.0% | 22 | 40.7% | |
| | Right | 7 | 43.8% | 11 | 47.8% | 0.171 | 21 | 42.0% | 28 | 51.9% | 0.064 |
| | Both | 4 | 25.0% | 1 | 4.3% | | 12 | 24.0% | 4 | 7.4% | |

Fisher's exact test, * *p* < 0.05; N (%) represents the number of participants whose foot conditions deviated from normal. A hallux valgus angle (HVA) ≤ 15° was considered normal. The number of floating toes, HVA, and distribution of weight were measured using the Foot Look (Foot Look Inc.).

Table 6 shows the sensitivity and vascular condition of the feet. When the examiner touched the participants' feet and asked whether they felt the stimulus, over 92.6% of men and women answered that they felt the touch sensation, although some had diabetes. When the examiner felt for a pulse at the posterior tibial pulse point, its prevalence in men was found to be lower at the support level than at the care level, while the opposite was true for women.

**Table 5.** Foot problems of participants with diabetes.

|  |  | n | % |
|---|---|---|---|
| Right_floating toe | (n = 26) | 19 | 73.1% |
| Right_toe deformities | (n = 26) | 21 | 80.8% |
| Right_skin lesions (corns and calluses) | (n = 26) | 3 | 11.5% |
| Right_arch deformities | (n = 26) | 22 | 84.6% |
| Right_thickend nails | (n = 26) | 16 | 61.5% |
| Right_ingrown nails | (n = 26) | 10 | 38.5% |
| Right_suspected or existing nail fungal infection | (n = 26) | 25 | 96.2% |
| Left_floating toe | (n = 26) | 18 | 69.2% |
| Left_toe deformities | (n = 26) | 19 | 73.1% |
| Left_skin lesions (corns and calluses) | (n = 26) | 5 | 19.2% |
| Left_arch deformities | (n = 26) | 20 | 76.9% |
| Left_thickend nails | (n = 26) | 17 | 65.4% |
| Left_ingrown nails | (n = 26) | 9 | 34.6% |
| Left_suspected or existing nail fungal infection | (n = 26) | 25 | 96.2% |
| Right_HVA $\leq 15°$ | (n = 25) | 19 | 76.0% |
| Right_$15° <$ HVA $\leq 20°$ | (n = 25) | 4 | 16.0% |
| Right_$20° <$ HVA $\leq 40°$ | (n = 25) | 2 | 8.0% |
| Right_HVA $> 40°$ | (n = 25) | 0 | 0.0% |
| Left_HVA $\leq 15°$ | (n = 23) | 17 | 73.9% |
| Left_$15° <$ HVA $\leq 20°$ | (n = 23) | 3 | 13.0% |
| Left_$20° <$ HVA $\leq 40°$ | (n = 23) | 3 | 13.0% |
| Left_HVA $> 40°$ | (n = 23) | 0 | 0.0% |

**Table 6.** Sensitivity and circulation assessment for participants.

| | Men | | | | | Women | | | | |
|---|---|---|---|---|---|---|---|---|---|---|
| | Support Level (n = 17) | | Care Level (n = 27) | | | Support Level (n = 53) | | Care Level (n = 63) | | |
| | n | Rate | n | Rate | *p*-Value | n | Rate | n | Rate | *p*-Value |
| Right foot | | | | | | | | | | |
| Sensitivity of the first toe | 16 | 94.1% | 25 | 92.6% | 1.000 | 52 | 98.1% | 62 | 98.4% | 1.000 |
| Sensitivity of the third toe | 16 | 94.1% | 25 | 92.6% | 1.000 | 52 | 98.1% | 62 | 98.4% | 1.000 |
| Sensitivity of the fifth toe | 16 | 94.1% | 25 | 92.6% | 1.000 | 52 | 98.1% | 62 | 98.4% | 1.000 |
| Palpable posterior tibial arteries | 10 | 58.8% | 8 | 29.6% | 0.068 | 25 | 47.2% | 36 | 57.1% | 0.351 |
| Left foot | | | | | | | | | | |
| Sensitivity of the first toe | 16 | 94.1% | 25 | 92.6% | 1.000 | 53 | 100.0% | 63 | 100.0% | 1.000 |
| Sensitivity of the third toe | 16 | 94.1% | 25 | 92.6% | 1.000 | 53 | 100.0% | 63 | 100.0% | 1.000 |
| Sensitivity of the fifth toe | 16 | 94.1% | 25 | 92.6% | 1.000 | 53 | 100.0% | 63 | 100.0% | 1.000 |
| Palpable posterior tibial arteries | 9 | 52.9% | 11 | 40.7% | 0.539 | 26 | 49.1% | 32 | 50.8% | 1.000 |

N (%) represents the number of participants who answered "I feel it" when an examiner touched their toe (sensitivity). N (%) represents the number of participants whose pulse could be palpated by the examiner.

Table 7 presents the results of the multiple regression analysis. Right toe grip strength had a statistically significant association with arch deformity ($p = 0.012$), suspected or existing nail fungal infection ($p = 0.034$), and nail thickness of the right foot ($p = 0.040$). Table 7 shows that corns, calluses, and toe deformities of the right-sided foot were significantly associated with the results of the walking test ($p = 0.026$ and $0.033$, respectively). No significant association was found between left-sided toe grip strength and left-sided foot-related items. Figure 4 illustrates the logical tree obtained from the regression analysis (Table 7).

**Table 7.** Multiple regression analysis: the association of toe grip strength and walking with foot parameters.

| Item | Dependent Variable | Independent Variable | Partial Regression Coefficient | 95% Confidence Interval | | Standardized Partial Regression Coefficient | *p*-Value |
|---|---|---|---|---|---|---|---|
| | | | | Lower Limit | Upper Limit | | |
| Toe grip strength | Right toe grip strength (n = 136) | Constant term | 8.97 | 6.36 | 11.58 | | <0.001 ** |
| | | Right arch deformity | −1.87 | −3.32 | −0.42 | −0.21 | 0.012 * |
| | | Right suspected or existing fungal infection of nails | −2.52 | −4.86 | −0.19 | −0.17 | 0.034 * |
| | | Right nail thickness | −1.14 | −2.23 | −0.06 | −0.17 | 0.040 * |
| | Left toe grip strength (n = 135) | Selected independent variable | - | - | - | - | - |
| Walking speed | Walking test (s) (n = 119) | Constant value | 5.16 | 3.79 | 6.52 | | <0.001 ** |
| | | Right corns and calluses | 1.89 | 0.23 | 3.56 | 0.20 | 0.026 * |
| | | Right toe deformity | 1.63 | 0.13 | 3.13 | 0.19 | 0.033 * |

* $p < 0.05$; ** $p < 0.01$; stepwise method (input: $p < 0.05$, output: $p > 0.1$); independent variables to enter (for both sides): toe spreading width, floating toes, toe deformities, corns and calluses, arch deformities, nail thickness, ingrown nails, suspected or existing fungal infection of nails, and weight distribution.

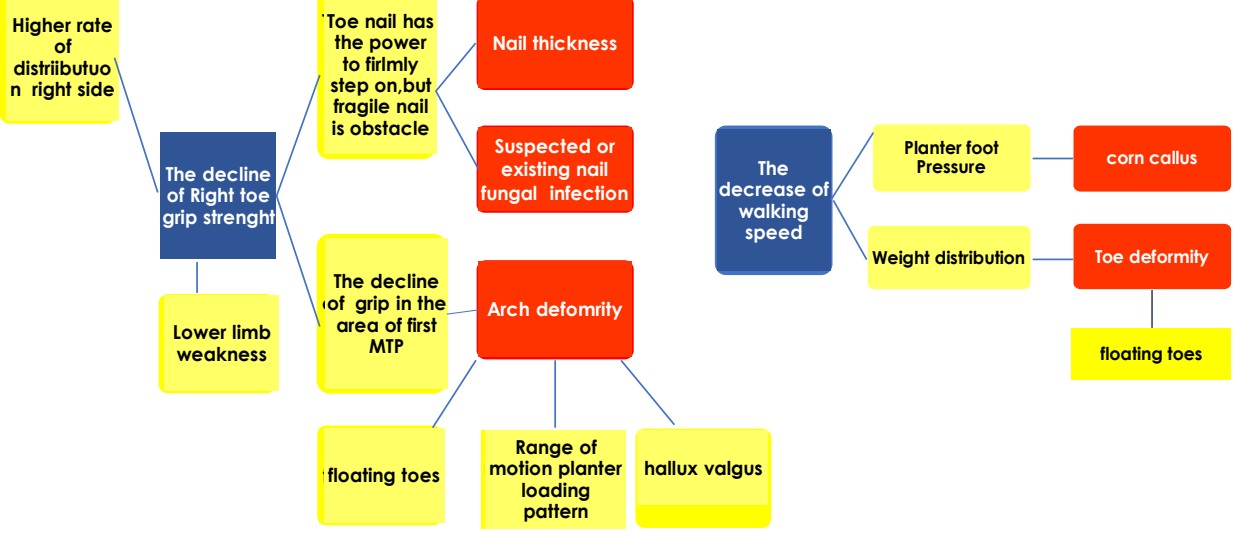

**Figure 4.** Logic tree obtained from the regression analysis of Table 7.

Since there are many measurement items, it was necessary to break down the factors and organize them in a hierarchy. Selected items that may be associated with toe grip strength and walking speed were input for analysis.

The blue box represents the dependent variable. Yellow boxes are related factors which cause the item of the independent variables. Red boxes represent independent variables.

Figures 5 and 6 show the scatter plot indicating the association between toe grip strength and waling speed. As Table 8 indicated, there was poor association between right toe grip strength and walking speed (R = −0.197). There was fair association between left toe grip strength and walking speed (R = −0.329).

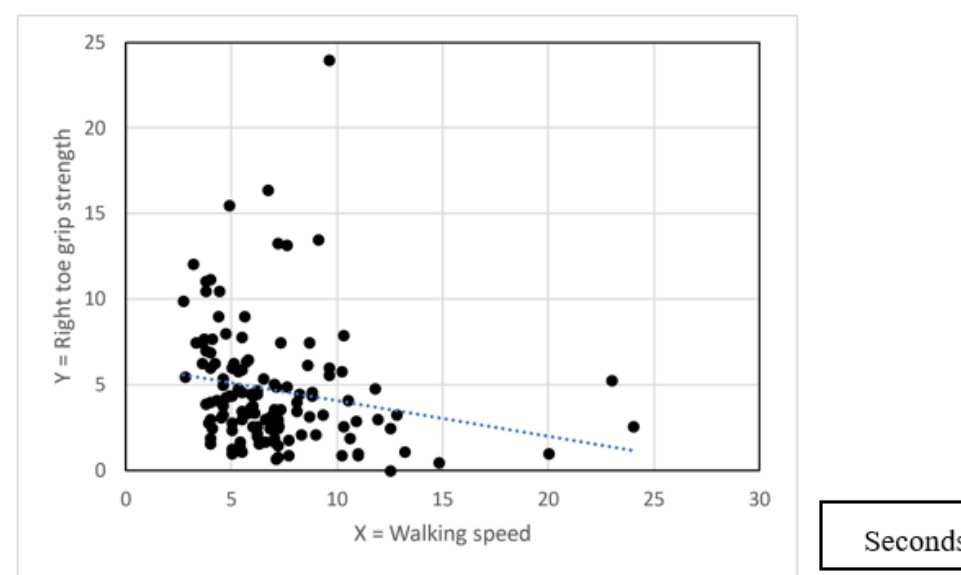

**Figure 5.** Association between right toe grip strength and walking speed.

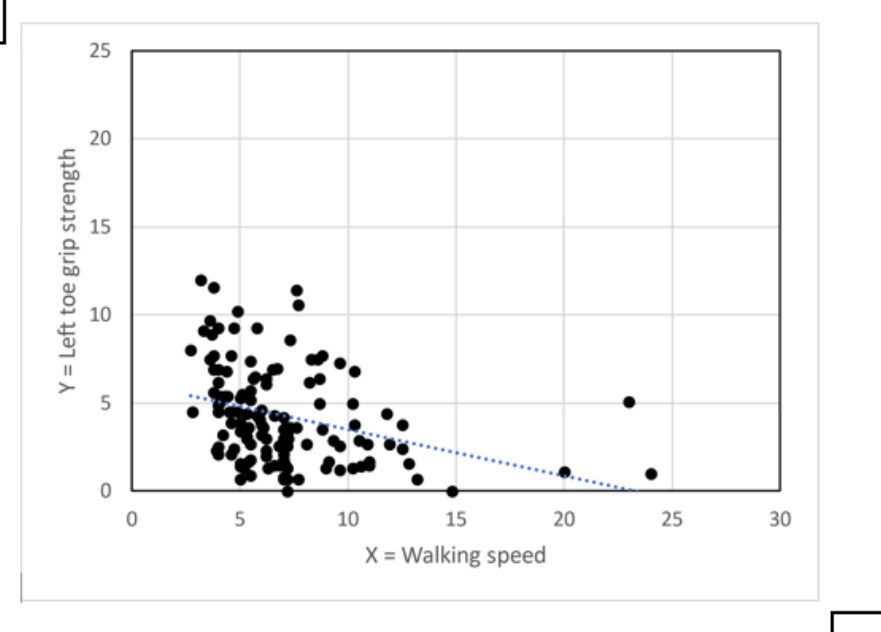

**Figure 6.** Association between left toe grip strength and walking speed.

**Table 8.** Association between toe grip strength and walking speed.

|  | | Walking Test (s) | |
| --- | --- | --- | --- |
|  | **n** | **r** | ***p*-Value** |
| R toe grip strength | 131 | −0.197 | 0.024 * |
| L toe grip strength | 131 | −0.329 | 0.000 * |

* $p < 0.05$; Pearson correlation; the walking test measured a 4 m walking speed; toe grip strength was measured using a toe grip dynamometer (T.K.K.3364 Takei Scientific Instrument).

## 4. Discussion

Older people have many foot problems. Moreover, foot-related problems were significantly associated with toe grip strength and walking speed.

### 4.1. Prevalence of Foot Conditions in Older People

Our study target was older people in the community who use long-term care insurance. Various bodily factors may negatively affect the foot health of older people with aging. Our investigation of foot items indicated that the incidence rates of floating toes, toe deformities, nail color change, arch deformities, nail length, thickened nail, skin dryness, edema, skin color change, and suspected or existing nail fungal infection were relatively high, whereas the prevalence rates of corns and calluses were low. Physical condition varies among community-dwelling older people; hence, foot conditions also differ among them. The participants in this study predominantly had long-term illnesses, such as cerebrovascular disease and diabetes, which are well known to limit ADLs and walking performance. Diabetes may cause peripheral neuropathy and accelerate foot-related issues that could trigger other foot disorders.

The reason for the lower prevalence rates of corns and calluses may be that the older population spends less time walking; therefore, the repeated pressure on the sole is minimal.

The low rate of palpable pulse points on the posterior tibial arteries, as well as the low temperature of the lower extremities and the coldness of feet observed when the examiner touched the skin, may also indicate reduced foot circulation because of aging, sedentary behavior, and lack of foot movement. Although we did not use Doppler and the results may not accurately detect the pulses, palpable pulses were weak among participants. The change in the condition of the peripheral blood vessels with aging is also related to nail color and nail growth [59,60].

A high incidence rate of foot skin dryness was present in this study, which supports previous findings that anhidrotic skin was prevalent in older people. This highlights the need for skincare among others [39]. A few studies reported that 44–45% of older people receiving in-home care [6] and 52.6% of residents of nursing homes had dry skin [61]. Both caregivers and older people should understand the mechanism of the skin barrier and apply the appropriate skin-protecting ointment.

In this study, we assessed possible nail fungal infections on the basis of observations. Nails with fungal infection become friable and appear yellowish [62]. Nail conditions can be detected through mycological tests, but the false-negative culture rate of this test is at least 30% [63]. Foot fungal infection is an indicator of lower limb cellulitis development [45]. Additionally, it may lead to psychological disorders due to its appearance [64]. Subjects with more deformities on the right side could likely maintain their right foot longer on the floor and apply a higher pressure on it due to lower mobility and fewer motor skills on the right side.

### 4.2. Foot Item Variables Correlate with Toe Grip Strength and Walking Speed

Previous literature reported reference values for toe grip strength targeting for the adults (20–79 years) [65]. Based on the literature, we predicted the normal value of toe grip strength of older adults.

No significant association was found between left-sided toe grip strength and left-sided foot-related items. This finding may be associated with a higher rate of participants represented by right weight distribution to the floor.

The two examiners looked at the image of the foot on screen and discussed which foot tended to press more firmly on the floor or whether both feet had equal pressure. More participants tended to favor their right foot over their left. The data were probably collected more accurately on the right foot because it was firmly planted on the floor. However, more studies are needed to explore this finding. As the study results indicated, toe and arch deformities were associated with the toe grip force of the right foot.

As the bar of the device measuring toe grip strength is gripped at the first metatarsophalangeal (MTP) joint of the foot, an abnormality in the foot arch may lead to a decrease in toe grip strength. However, more data need to be accumulated on the relationship between toe grip strength and arch correction.

The factors for the development of arch deformity are interlinked with muscle degeneration, age-related weakening of the tendons and ligaments, lifestyle, walking, shoe selection, and reduced activity. Arch deformities cause a decline in the range of motion, a change in the plantar loading pattern, and postural stability [66,67], causing impaired balance due to floating toes. Aging can be considered a key factor in changing foot shapes [68–70] and declining toe grip strength [18,71].

The foot arch is known to be associated with the plantar fascia, which is also known as the plantar aponeurosis [72]. In one study, Erdemir et al. [73] described the role of the plantar fascia during walking. The study, which evaluated the association between the plantar fascia and toe force, was conducted on 20 people and found that there was no relationship between the progressing condition of the first MTP joint and the presence of plantar fascia. This study concluded that further investigation on this matter would be required [74]. In a study, Digiovanni et al. [75] stated that adequate rehabilitation affects the stretching of the plantar fascia. In another study, Mickle et al. [76] suggested "plantar fascia stretching, perception, and balance training, along with pain management and weight control" as components of a rehabilitation program to prevent falling. However, we suggest that the content of an adequate rehabilitation program should be further assessed. For example, instead of merely exercising the muscles, stretching the foot before exercise may prevent pain generation and could make the exercise more effective. Additionally, various auxiliary tools and pads are available for arch correction; thus, further data should be gathered to verify that they have substantial effects on toe grip strength.

There are several possible reasons for the association between toe grip strength and nail problems. The association between nail problems and toe force was previously reported [19, 77,78]; however, the number of studies on this topic is limited in Japan and overseas. Abnormal nails, including thickened, deformed, or friable nails caused by a fungal infection, may trigger a decline in toe grip strength because the nails may press against the floor as the toes are flexed. Thus, nail care significantly affects toe grip strength.

Walking speed was significantly associated with the existence of corns and calluses as well as toe deformities. The formation of corns and calluses themselves and pain caused by them may be associated with plantar pressure, leading to a decline in walking speed. The association between walking speed and plantar pressure has been widely studied [70,79–81].

Another study reported the effectiveness of treatment in resolving pain caused by keratosis, such as corns and calluses [82,83]. However, the association of corns and calluses with plantar pressure remains unclear [84,85]. Another study revealed that plantar pressure is increased in older people with callused regions, and an intervention study indicated that the removal of calluses reduces plantar foot pressure [86]. The lower prevalence of calluses and corns indicated that the walking time of the study participants was reduced due to aging. One should avoid the assumption that corns and calluses affect toe grip strength because the generation of corns and calluses is limited for this targeted population, leading to a low prevalence.

Toe deformities include hallux valgus and lesser toe deformities. Some patients have no toe deformity, whereas others have multiple toe deformities. Mickle et al. [79] suggest that lower toe deformation changes the load distribution during walking because such a toe condition results in the toes being pulled back into extension, thus reducing their contact area.

As the presence of corns and calluses triggers pain, both toe hallux valgus and lesser toe deformities may generate pain [82]. This pain may greatly affect walking speed.

Hughes noted that toes are in contact with the ground for three-quarters of the walking cycle and fulfill a weight-bearing function [87]. The balance transition from the medial to

the lateral side of a patient with hallux valgus is significantly delayed [88]. The toe plays an important role in maintaining the body's balance and reflects lower limb function more broadly. Therefore, appropriate efforts to strengthen toe function, such as washing, nail care, stretching, and exercise, are required. In the present study, no significant relationship was observed between toe grip strength or walking speed and hallux valgus or floating toes.

When the foot arch becomes deformed, the occurrence of a bunion may be inevitable. When the foot spreads laterally due to muscle degeneration or other reasons, such as arch deformation, it causes the adductor muscle of the toes to be pulled outward, thereby leading to the formation of hallux valgus. Many studies have primarily focused on hallux valgus, among other toe deformities [56,89]. Furthermore, a report indicated an association between the degree of hallux valgus and toe grip strength [90].

In this study, the analysis was conducted by dividing the degree of hallux valgus into three stages. Our study revealed that the prevalence of hallux valgus (>15°) in women was 29.9–35.9%, which was higher than the prevalence in men; however, it did not reveal an association between the angle of hallux valgus and walking speed. The small sample size may be associated with this apparent lack of connection. Future studies with more samples may explore the association between the angles of hallux valgus and walking speeds.

An arch deformity may prevent the toes from firmly attaching to the floor, resulting in floating toe complications, which have recently gained attention from researchers [55,91]. However, the number of studies remains limited both in Japan and overseas. Uritani et al. [85] concluded from their study that floating toes affect walking; however, further studies are required to confirm this. Since toe grip force is significantly related to several foot problems shown in Table 7, the left-side foot may be fairly associated with walking speed without less impediments of the foot. A previous study reported an association between decreased toe flexor strength and slow walking [92]. Further studies will explore this association in stratified age groups.

The data obtained by accessing and measuring various aspects of the foot revealed the prevalence of foot disorders in aged people. The results of multiple regression analysis implied the presence of specific impairments in foot condition, which may reduce walking speed and toe grip force.

These findings provide insights into potential future strategies to prevent foot conditions from worsening. Podiatric care with effective nail care, prevention of skin lesions, and adjustment of arch deformities could help patients maintain their toe grip strength or walking speed, which are key indicators of foot health. Skincare also helps keep the feet in good condition by preventing cracks in the skin.

Older people face a high risk of falling and becoming bedridden. According to the Ministry of Health, Labor and Welfare, fractures and falls are the third most common cause of the need for long-term care in people who are certified as requiring high-level care (level 4 or 5) through long-term care insurance [93]. Therefore, new approaches to incorporate appropriate foot care into daily practice as part of day services or daycare would be desirable. Information on how to prevent the occurrence of nail thickening, nail fungal infection, toe and foot deformity, corns, and calluses as well as information on how to appropriately provide foot care and promote plantar fascia stretching and foot exercise would be key components of future strategies for preserving foot health among community-dwelling people in this as well as other age groups. Additionally, the delivery of knowledge and the practice of foot care among the caregivers for people of this age group would be significantly helpful for preserving their foot health.

### 4.3. Limitations

Our study has limitations with regard to the data collected. Although there are very few studies that describe the factors causing deformation of the arch and toes, there are various possible factors that need to be considered. This study, however, does not mention them. Additionally, the increased association of the decrease in skeletal muscle mass with

arch and toe deformity because of aging was not evaluated in this study, although this topic has attracted a great deal of attention in recent years.

Furthermore, fall injuries were initially included in the data collection but were eventually eliminated because the information provided by the participants and the staff was uncertain. Physical activity was initially measured using a short battery of physical performance tests, including balance, chair standing, and walking tests, most of which were subsequently cut. Performing balance and chair standing tests made the participants, as well as the staff, extremely anxious because of mechanical body changes, such as heart rate elevation and instability during standing. Therefore, we selected only the 4 m walking test.

### 5. Conclusions

In summary, the study shows that the rate and specific elements of foot impairment in older people may be associated with walking speed and toe grip strength. This finding provides useful insight into the foot conditions of older people. Further studies should be conducted to target more people in this age group and investigate strategies to prevent falls and bedridden conditions by incorporating foot care into routine practice at all facilities; however, many existing obstacles need to be overcome. The importance of foot and foot care education needs to be promoted in Japan and its knowledge and skills need to be delivered to nurses and care workers. For our future research, consistency in the time and target site of temperature measurement, a larger number of participants, a larger number of replicate measurements per patient, and a comparison of the data with those of younger people would improve accuracy.

**Author Contributions:** Conceptualization—K.F., A.M. and T.K.; Data curation—K.F., N.K., R.N. and Y.S.; Formal analysis—K.F. and T.F.; Funding acquisition—K.F.; Investigation—K.F., N.K., R.N. and Y.S.; Methodology—K.F., T.F., T.K. and M.S.; Project administration—K.F.; Resources—K.F.; Supervision—K.F.; Validation—K.F.; Visualization—K.F.; Roles/Writing—original draft—K.F., T.K. and N.K. All authors have read and agreed to the published version of the manuscript.

**Funding:** The questionnaire development and data collection and analysis were funded by Japan Society for the Promotion of Science (NO: 19K11111).

**Institutional Review Board Statement:** The research conformed to the Declaration of Helsinki, 2013. The study and informed consent were further approved by the ethics committee of Nagoya University (approval number: 2019–0150).

**Informed Consent Statement:** We obtained written and oral informed consent from all participants with the approval of facility providers and some of the families.

**Data Availability Statement:** The datasets used and/or analyzed during the current study are available from the corresponding author on reasonable request.

**Acknowledgments:** K.F. wishes to acknowledge the work and support of the following: All centers, nurses, and care workers who participated in and shared their precious time for this study; I. Yamamichi, Fusspfleger at the Japan Foot-Care Fusspfleger Schoo; R Fukushima for his expertise in statistics; and N. Tsutumi, M. Kawabata, W. Nakamura, H. Taketani, S. Watanabe, K. Suzuki, A. Yamada, R. Nakamura, and I. Honda for their valuable assistance for this research.

**Conflicts of Interest:** The authors declare no conflict of interest.

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
