# Peer review of "Foot Problems and Their Associations with Toe Grip Strength and Walking Speed in Community-Dwelling Older Individuals Using Day Services: A Cross-Sectional Study"

_nursrep, doi:10.3390/nursrep13020062_

Round 1

Reviewer 1 Report

I thank the authors for exploring an important area of foot disorders research. This research aimed to determine the prevalence of foot disorders and examine the associations between foot conditions and toe grip strength, toe force, and walking speed among people aged over 65 years in the community in Japan. The prevalence rates of skin dryness and suspected and existing fungal infections in nails were found to be high among older individuals in this study. This study also highlighted that right-sided foot-related problems were significantly associated with right-toe grip strength and walking speed. Overall, this manuscript conveys the main message clearly. However, for publication, the following issues need to be considered.

Overall, the authors should improve its readability by considering the fact article will be read by many international readers. I would also appreciate it if the manuscript could be submitted with line numbers to make the review process smooth from both sides.

Abstract

1.     The second line is incomplete. Please rephrase.

2.     Authors should clearly mention sample size, year, study settings, age group, and location, including country.

3.     Was the final sample size 176? The result section says 160.

4.     “The prevalence rates of skin dryness and suspected and existing fungal infections in nails were high in both sexes regardless of the level of care required.”—A numeric figure for the prevalence would be helpful.

Materials and Methods

5.     Page 2: Design-> “who attended daycare or day rehabilitation centers.” where? Was it all over Japan or in particular locations?

6.     Page 3, paragraph 2: I  would strongly recommend providing calculation details in a separate supplement file. Please also check if the figure is correct (for example, power=..). Please also provide the full forms of f2 and R2.

7.     Authors need to provide a proper scientific reference for sample size calculation. Reference 33 is not a proper reference.

8.     Page 7, Data analysis: Statistical analysis would be a more suitable sub-title. Authors need to mention why did they use Fisher’s exact test and Mann-Whitney U test.

9.     The description of the dependent variable of the multiple regression needs to be presented under a separate sub-section.

10.  Multiple regression as a method of analysis is not mentioned in “Data analysis” section.

Results

11.  Page 7, results (1st paragraph): Authors mentioned “some patients”, “some participants” etc. Please provide numeric figures.

12.  Page 8” “The demographic characteristics of the participants, including age, Barthel Index, and history of the disease, are presented in Table 2.”—I do not see any age distribution in the table. Moreover, facility location is an important demographic factor that should be included/ presented in the table.

13.  “The most prevalent diseases were hypertension (41.9%), brain disease (history of cerebrovascular accidents) (19.4%), and diabetes (16.3%).”-  which table?

14.  “The results of the 4-meter walk test, skin temperature, toe grip force, and toe spread-ing width are shown in Table 3.” – Is the table missing?

15.  “Table 5 presents the foot problems of participants with diabetes.”- No results were discussed.

16.  For the overall result section, please consider mentioning the figure presented in the table. If the figure is not shown in the table, please show how it was derived from the table in brackets.

Reviewer 2 Report

The article entitled “Foot problems and their associations with toe grip strength and walking speed in community-dwelling older individuals using day services: A cross-sectional study” is interesting and focuses on foot problems that affect the quality of life in elderly people. However, it is difficult to follow and suffers from a series of technical flaws.

First, the authors need to clarify if the toe grip strength was normalized and how it was done. Most probably the age and weight of the participants will have an effect on the data.   

The number of lines in the manuscript should be included, this will make it easier to comment on and identify the suggested changes.

When the authors commented “A previous study used these devices to investigate the association…”, the authors need to mention the name of the devices or technical features of them.

It is not clear that the Foot Look device measures plantar foot pressure as indicated by the authors, this needs to be clarified.

The authors need to clarify the difference between toe grip strength and toe force as they are mentioned in the second objective at the end of the introduction section.

The authors need to clarify why they are having two different sample populations and why they decided to go for the sample obtained by G*Power. Furthermore, there are some typo errors that need to be corrected. 

Which are the three dependent variables? This needs further explanation. 

Why did the authors decide to include participants with and without walking aid? This will have an impact on the walking velocity and the comparison among them is going to be difficult.  

The authors need to think if considering 20 facilities out of 400 is representative.

If the authors are trying to measure the foot type by measuring the soles of the feet with the 2D scanner, they should reference an article instead of a book [52].

Revise the references within the text, the number should be in the order they were included.

It is not clear how the Foot look device measures the Hallux Valgus. The authors included the reference of Coughlin MJ [84], but Coughlin used radiographic images. The authors need to clarify how they measured the Hallux Valgus and provide evidence of the accuracy of the results.

Revise the measurement unit of toe grip strength, kg.

Revise all web pages included in the manuscript as some of them are no longer available. https://www.takei-si.co.jp/productinfo/detail/268.html

The demographic characteristics of the patients need to be included.

The title of Table 2 is not related to the content. Furthermore, the demographic characteristics of the participants are not in Table 2 as commented within the text.

Table 3 does not mention the 4-meter walk test, and so on. The title of Table 3 is wrong. Furthermore, the notes below the table are not related to the content of the table.     

Where is the information on the 4-meter walk test, skin temperature, toe grip force, and toe spreading width?

There are some typo errors in Figure 4. Moreover, how was the pain measured?

How do the authors validate the information of the diseases, Diabetes, hypertension, kidney disease, heart disease, lung disease, so on? Was a test of diagnosis performed or requested to the patients?

A detailed explanation of Figures 5 and 6 needs to be included. The measurement units in Figure 5 need to be included in both axes. The same for Figure 6.

The second paragraph of the limitation section is not a limitation of the study.

The sentence “Walking speed and toe grip strength were significantly associated in this study” cannot be supported by the results found in the study. 

Reviewer 3 Report

Dear Authors, excellent work! It was an excellent opportunity to read your work. I made some nites on the pdf of the paper that I send in attachment.

Take my comments as positive, and decide if you want to change

1. Abstract

Statement of the Aim of the study - To determine the prevalence of foot problems and their associations with toe grip strength and walking speed in frail older people. 

2. Introduction

A lot of small sentences should be integrated in one.  Example: "Frailty is defined in various literature [9,10,11]. According to a statement from the Japan Geriatrics Society, the concept of frail is “a state of increased vulnerability to stress. A state of being prone to turning points such as life dysfunction, need for nursing care, and death” [12].

3. Material and Methods

Date of collection - Data were collected from July to October in 2019 - These data as almost 3 years

The date of the paper appears 2021? - Nurs. Rep. 2021, 11, FOR PEER REVIEW???

The sample size calculation is confusing, data related with epidemiology is from 2022 and the study was performed in 2019? The Sampling should be more clear, try to rewrite if possible.

I suggest that the images 1 and 2 should appear following the text and respective reference, they should appear after the table at the description of each instrument used, not in the beginning.

4. Results

Consider to change the tables, they have a lot of information not needed in the end. Use only the necessary itens described.

Use a small introduction before showing the results in the table, for example - Table 5 presents the foot problems of participants with diabetes without any description of the table (what is relevant to be seen?)

Table 7 and 8 are presented simultaneously in the same paragraph and the tables are presented one attached to the other. I suggest to make a description or comment between them.

Reflect on the need of some may tables that can be as appendices

Figure 4, 5 and 6 are presented without description. Are they figures or graphs? Missing information of interpretation. Please rewrite these descriptions. 

Discussion should explore more the regression model - Logic tree obtained from the regression analysis. 

Conclusions are interesting. Felt the need for the implications to practice - future interventions in clinical approach, because the implications to research are written 

Round 2

Reviewer 2 Report

Further explanation needs to be done to clarify the huge difference in the sample calculation 385 vs 68 and 119. That part is confusing.

Revise all the references and place them in the order that they appear. Furthermore, revise all the references, for example, reference [84] is not related to the HVA.

In page 7, when the authors said “From the image, the toes touching the floor can be observed, …” which image do you refer to?

Revise this phrase “Additionally, since men are women are different in terms of muscle mass…”

A table caption needs to be included (Table 2).

Revise the format of the information below table 2, page 9. It seems wrongly placed.

Further explanation needs to be done to clarify the measurement of the hallux valgus, it seems that the Foot Look uses some bony landmarks of the foot but the hallux valgus is related to the longitudinal orientation of the phalanges and metatarsal bones.

Revise Figure 4, correct the phrase “planter pressure”

Revise Figures 5 and 6, labels are wrong and misplaced.

Revise the format of the references. It has to be the same for all of them.

This sentence is incomplete “Figures 5 and 6 show the scatter plot indicating the association”, the association of what?

Why the walking speeds in figures 5 and 6 are different?

Revise the format of the whole document as in some sections the font size is different.  

There are still some typo errors in the document. Spell check required.

Author Response

To reviewer 2.

Thank you so much for review for my paper again. I have changed as follow .

Further explanation needs to be done to clarify the huge difference in the sample calculation 385 vs 68 and 119. That part is confusing.

→I added explanation. I also revised the sentences P7 123-124. I stated that 119 was satisfied the required number of participants, but 119 was the results of the number to analyze the association between walking and selected foot items. Therefore, I revised. 160 instead of 119 based on the Fig 3. 

Revise all the references and place them in the order that they appear. Furthermore, revise all the references, for example, reference [84] is not related to the HVA.

→No 84 is about The Ipswich Touch Test. As you see P38, the citations of 83, 84 is about Ipswich test, however, I added one citation ( was deleted before but I put is as reference again NO84) Citation number was changed to No85 and 86

In page 7, when the authors said “From the image, the toes touching the floor can be observed, …” which image do you refer to?

→As shown in Fig 1, the toes touching the floor can be observed from the display of Foot Look

Revise this phrase “Additionally, since men are women are different in terms of muscle mass…”

→P11 112  I changed  to “Women and men have different muscle mass”.

A table caption needs to be included (Table 2).

→I added as Sample Characteristics

Revise the format of the information below table 2, page 9. It seems wrongly placed.

→I revised.

Further explanation needs to be done to clarify the measurement of the hallux valgus, it seems that the Foot Look uses some bony landmarks of the foot but the hallux valgus is related to the longitudinal orientation of the phalanges and metatarsal bones.

→I added more detailed explanation about FOOT LOOK

Revise Figure 4, correct the phrase “planter pressure”

→I changed to planter foot pressure instead of planter pressure. I also changed font and color of the Figure 4 to be readable.

Revise Figures 5 and 6, labels are wrong and misplaced.

→I have changed

Revise the format of the references. It has to be the same for all of them.

This sentence is incomplete “Figures 5 and 6 show the scatter plot indicating the association”, the association of what?

→I have changed

Why the walking speeds in figures 5 and 6 are different?

→I recalculated and revised it. I also changed Table 8 which is related to figure 5 and 6.

Revise the format of the whole document as in some sections the font size is different. 

→I and coauthors checked it again and reivised it.

There are still some typo errors in the document. Spell check required.

→I and coauthors checked it again and revised it.( I highlighted green)